# Catalytic Performance of CPM-200-In/Mg in the Cycloaddition of $CO_2$ and Epoxides

**Yunjang Gu, Youngson Choe and Dae-Won Park \***

Division of Chemical and Biomolecular Engineering, Pusan National University (PNU), Busan 46241, Korea; guyj1234@pusan.ac.kr (Y.G.); choe@pusan.ac.kr (Y.C.)
\* Correspondence: dwpark@pusan.ac.kr; Tel.: +82-51-510-2399; Fax: +82-51-510-8563

**Abstract:** Crystalline porous materials (CPM)-200-In and CPM-200-In/Mg metal-organic frameworks (MOFs) were synthesized by a solvothermal method and were characterized by using powder X-ray diffraction (PXRD), FT-IR, Brunauer–Emmett–Teller (BET), temperature programmed desorption (TPD), TGA, XPS, and SEM-EDS. They were used as heterogeneous catalysts for the cycloaddition of $CO_2$ with epoxides and found to be highly efficient toward the cycloaddition reaction at moderate reaction conditions under solvent-free conditions. The catalyst was easily separated by a simple filtration and can be reused up to five consecutive times without any considerable decrease of its initial activity. CPM-200-In/Mg showed excellent catalytic performance in the cycloaddition reaction due to the synergistic role of the acidic sites and basic sites. A plausible reaction mechanism for the CPM-200-In/Mg MOF catalyzed cycloaddition reaction is proposed based on the experimental results and our previously reported DFT (Density Functional Theory) studies.

**Keywords:** CPM-200-In/Mg; cycloaddition; $CO_2$; epoxide; cyclic carbonate





## 1. Introduction

The conversion of carbon dioxide to value-added chemical products has attracted scientists to develop efficient technologies for the reduction of $CO_2$ emission, which is a main cause of global warming [1–9]. The production of cyclic carbonates has been considered as a promising way to convert $CO_2$. Cyclic carbonates have a wide range of applications such as solvents, electrolytes, and intermediates for the synthesis of other monomers and pharmaceuticals [10–17]. Since $CO_2$ is a thermally and chemically stable molecule, various homogeneous (e.g., quaternary ammonium and phosphonium salts, Schiff bases, ionic liquids, alkali metal salts, and other organocatalysts) [18–23] and heterogeneous (e.g., metal oxides, functional polymers, and immobilized ionic liquids) [24–30] catalysts have been developed in the synthesis of cyclic carbonates. The heterogeneous catalysts have higher catalytic performance, easier separation, and higher reusability than homogeneous ones [27,28]. It is well reported that heterogeneous catalysts based on metal-organic frameworks (MOFs) can efficiently proceed the topic reaction (Scheme 1) [31–50]. Porous MOFs composed of metal ions and ligands have been widely used as catalysts for many chemical reactions due to their crystalline nature, tunable structure, highly specific surface area, good thermal stability, and easier mass transfer of reactants [45,51–53].

Recent studies on MOF-based materials for $CO_2$ mitigation are focused on the development of MOFs having high adsorption capacity and high catalytic performance at the same time [8]. Metal organic frameworks (MOFs), based on transition metals, are predominant over those based on main-group elements. However, it is reported that the MOF-74-Mg exhibited the highest $CO_2$ uptake capacity over its transition metal analogues such as MOF-74-Co and MOF-74-Ni [54]. The application of MOF-74-Mg as a catalyst for the cycloaddition of $CO_2$ and epoxides has also been reported previously [55,56]. Recently, bimetallic MOFs (Ni-Co and Co-Zn MOF) have been developed as catalysts in the cycloaddition reaction [57–59]. They have shown synergistic effects in promoting the

catalytic activity by increasing the number of acidic and basic sites. We also reported the synergistic role of Cu-Zr bimetallic MOF, which enhanced the number of basic sites for the cycloaddition reaction [59]. Zhai et al. [60] reported successful preparation of heterometallic MOFs, denoted as crystalline porous materials (CPM-200s) with systematic combinations of trivalent and divalent metals, and their high $CO_2$ adsorption capacity.

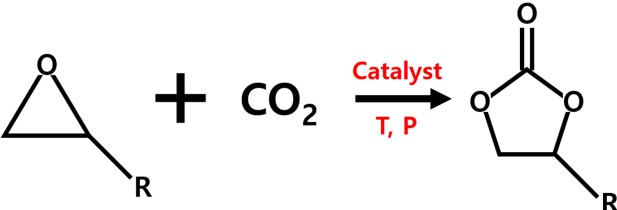

**Scheme 1.** Synthesis of cyclic carbonate from epoxide and $CO_2$.

In this work, therefore, we selected indium (In) as a second metal to Mg containing MOF and tried to suggest the promotional effect of the bimetallic In-Mg MOF in the cycloaddition reaction. We synthesized CPM-200-In/Mg MOF, and it was used as a catalyst for the first time in the cycloaddition reaction under solvent-free conditions. The bimetallic CPM-200-In/Mg showed better catalytic performance than CPM-200-In since the In-Mg bimetallic MOF exhibited higher $CO_2$ adsorption capacity, higher number of acidic and basic sites, and higher thermal stability. The CPM-200-In/Mg MOF catalyst can be reused up to five consecutive times without a considerable loss of its activity. A reaction mechanism is also proposed based on the experimental results and our previously reported DFT studies on the reaction steps, including reaction intermediates and transition states.

## 2. Results

### 2.1. Characterization of Catalysts

The powder X-ray diffraction (PXRD) patterns of the prepared CPM-200 catalysts were compared with the simulated single crystal pattern (Figure 1). The main characteristic peak corresponding to (200) face of CPM-200 catalysts in PXRD matched well with the simulated pattern of the Cambridge Crystallographic Data Center (CCDC) data, confirming the crystalline structure of the prepared catalysts.

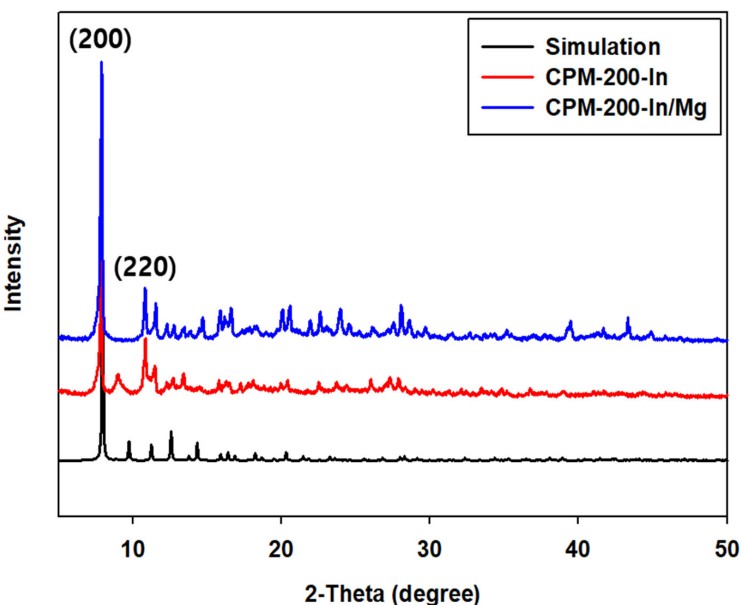

**Figure 1.** XRD patterns of crystalline porous materials (CPM)-200-In and CPM-200-In/Mg with simulated CPM-200 (Cambridge Crystallographic Data Center (CCDC) number: 1444375).

Figure 2 show SEM-EDS images of the synthesized CPM-200-In and CPM-200-In/Mg samples. The SEM images reveal that both CPM-200s have cubic like structures. As a result of EDS mapping, only In metal was observed in CPM-200-In, while In and Mg metals are well distributed in CPM-200-In/Mg.

The FT-IR spectra of the catalysts are shown in Figure 3. The broad peaks in the range of 3400–3450 cm$^{-1}$ are assigned to the O-H stretching frequency of coordinated water. The big peaks at 1635 and 1430 cm$^{-1}$ correspond to the symmetric and asymmetric stretching vibration modes of the COOH group. The symmetric stretching vibration of the In-O and Mg-O bond at 419 cm$^{-1}$ and 548 cm$^{-1}$ verifies the coordination of metals with the ligands [61,62].

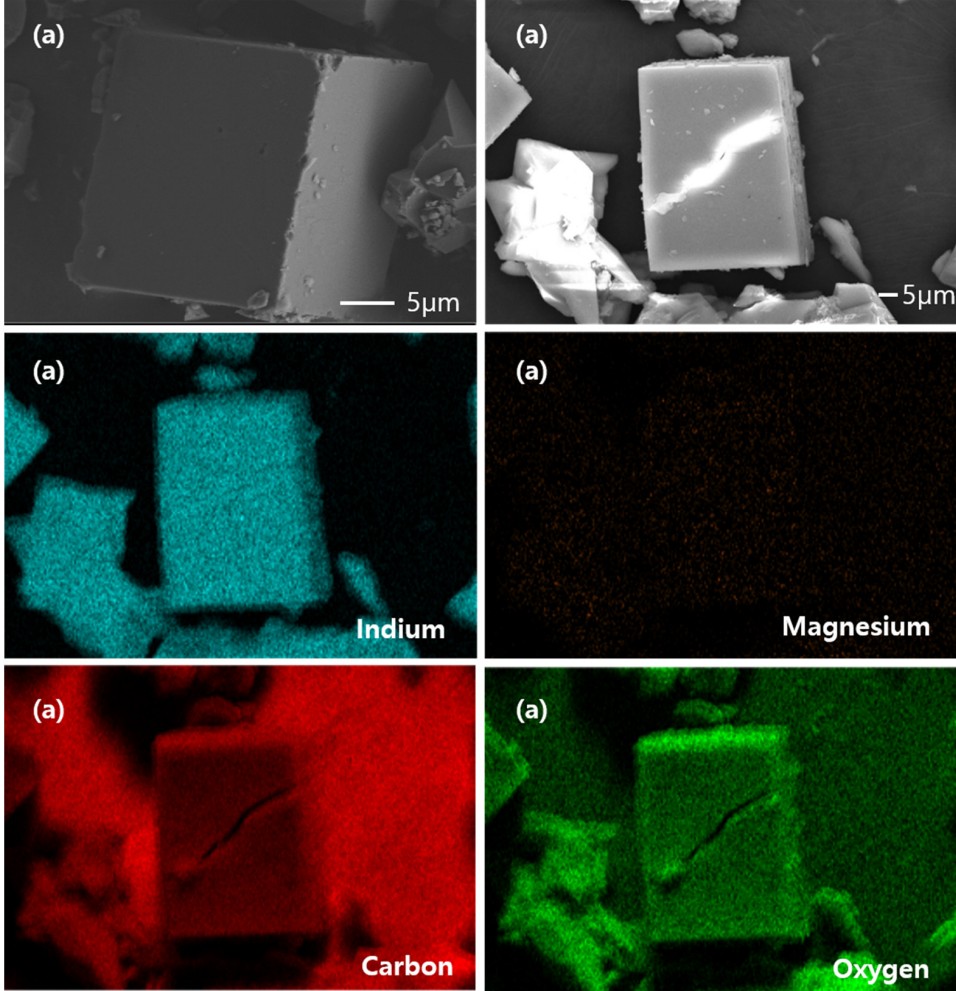

**Figure 2.** *Cont.*

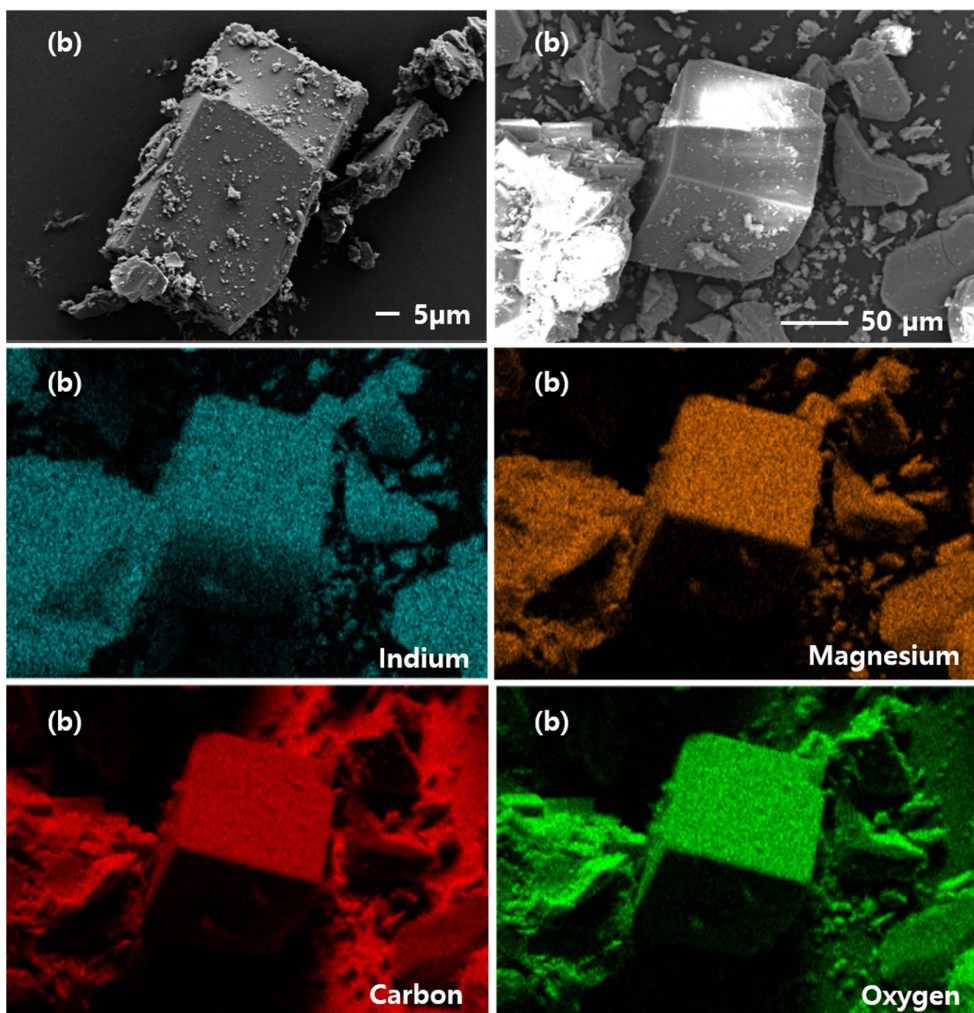

**Figure 2.** SEM-EDS image of synthesized catalysts: (**a**) CPM-200-In and (**b**) CPM-200-In/Mg.

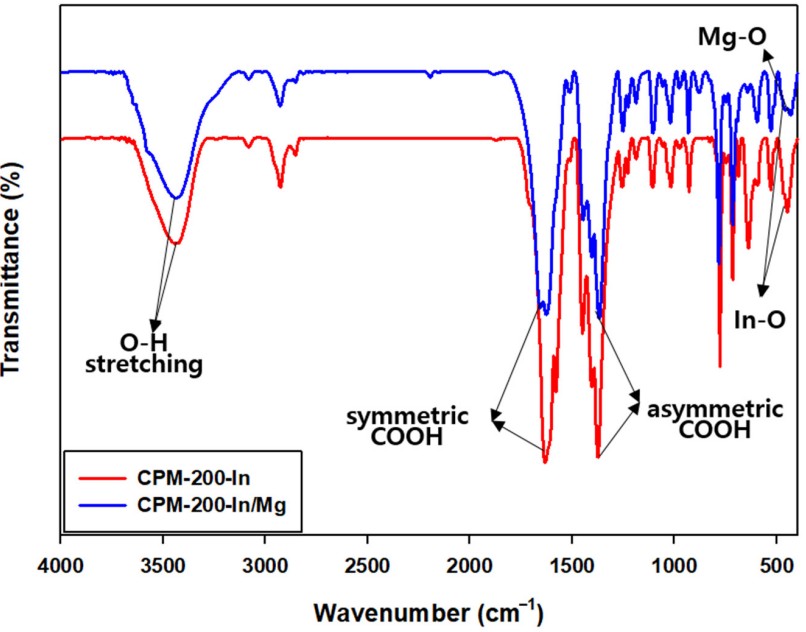

**Figure 3.** FT-IR spectrum of CPM-200-In and CPM-200-In/Mg.

TGA results of CPM-200-In and CPM-200-In/Mg (Figure 4) showed an initial weight loss of solvent inside their pores around 130 °C and 200 °C. Even after the initial weight loss, CPM-200-In/Mg exhibited better thermal stability than CPM-200-In, probably due to the better crystallinity from the synergy effect of two metals during CPM-200-In/M crystallization process [60] and/or to the smaller micropore diameter for CPM-200-In/M compared to CPM-200-In.

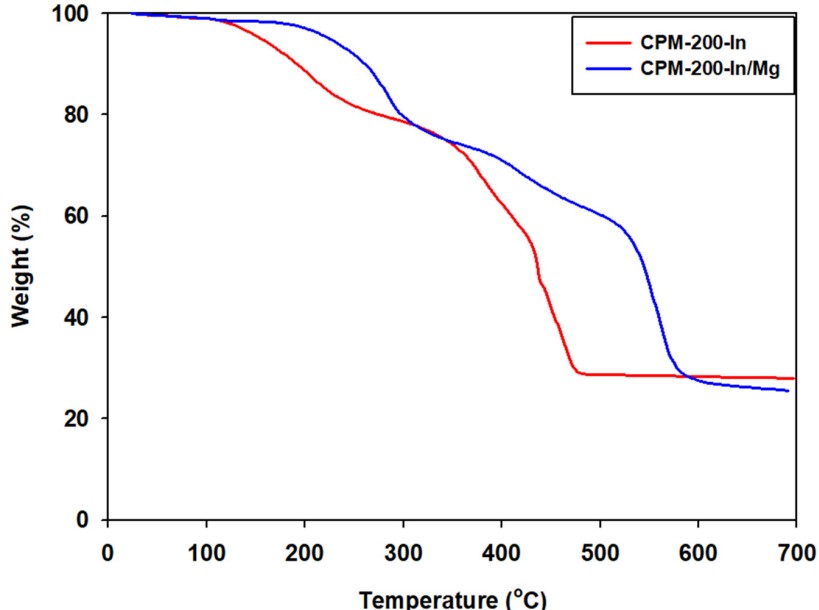

**Figure 4.** TGA curve of CPM-200-In and CPM-200-In/Mg.

To verify the elemental composition of both catalysts, elemental analysis (EA) and inductively coupled plasma–optical emission spectrometry (ICP-OES) were conducted, and the results in wt% are given in Table 1.

**Table 1.** Elemental analysis (EA) and inductively coupled plasma (ICP) results of CPM-200-In and CPM-200-In/Mg.

| Catalyst | Elemental Composition (wt%) | | | | | |
|---|---|---|---|---|---|---|
| | **In** | **Mg** | **C** | **H** | **O** | **N** |
| CPM-200-In | 35.84 | - | 30.05 | 1.86 | 26.69 | 4.81 |
| CPM-200-In/Mg | 13.13 | 8.20 | 36.89 | 2.96 | 31.68 | 7.26 |

X-ray photoelectron spectroscopy (XPS) was also performed (Figure 5) to verify the presence of bimetal in CPM-200-In/Mg. In 3d5 and Mg 1s peaks located at binding energies of 445.26 eV and 1304.30 eV show the presence of indium and magnesium metals in the MOF [63–65]. CPM-200-In also exhibits In 3d5 peak at the binding energy of 445.12 eV, demonstrating the presence of the indium atom.

The porosity and surface area of both CPMs were studied by the physical adsorption of $N_2$ at 77 K, and the type I adsorption isotherms revealed the existence of micropores (Figure 6). Brunauer–Emmett–Teller (BET) surface areas of 1231 $m^2g^{-1}$ and 756 $m^2g^{-1}$, and pore volume of 0.50 $cm^3g^{-1}$ and 0.44 $cm^3g^{-1}$, were calculated from $N_2$ adsorption isotherms for CPM-200-In/Mg and CPM-200-In, respectively. The pore size distribution curves confirm the presence of micropores in both catalysts, and the smaller micropore diameter of CPM-200-In/Mg than CPM-200-In. The presence of mesopores is known to be beneficial for promoting mass transfer [51–53]; however, no meaningful evidence of mesopore was observed in both CPM-200 catalysts.

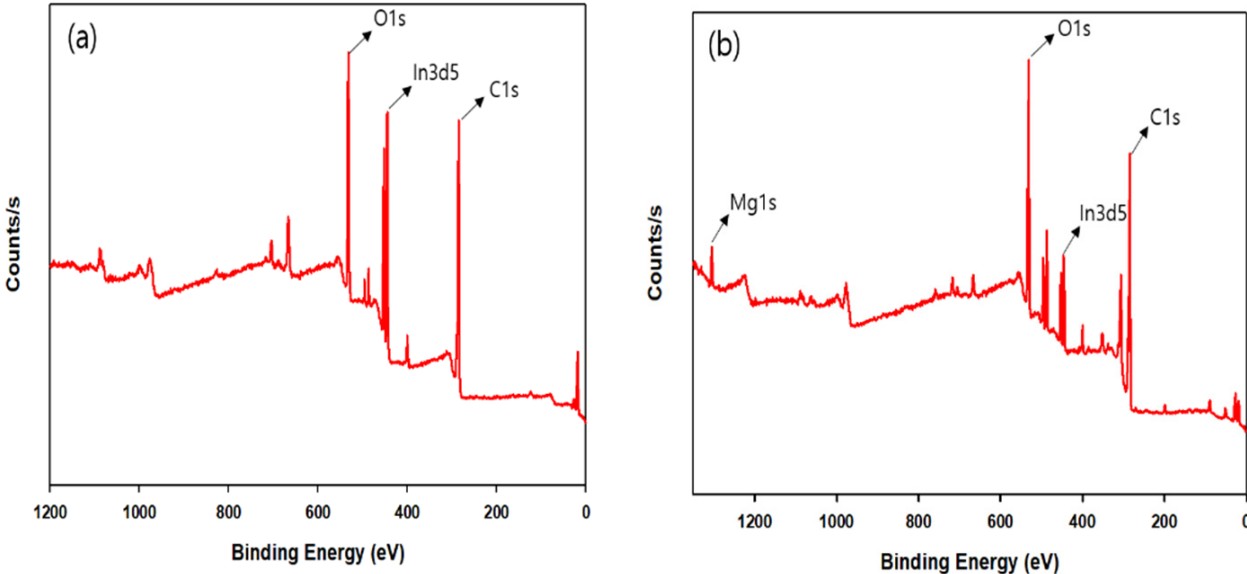

**Figure 5.** XPS spectrum of (**a**) CPM-200-In and (**b**) CPM-200-In/Mg.

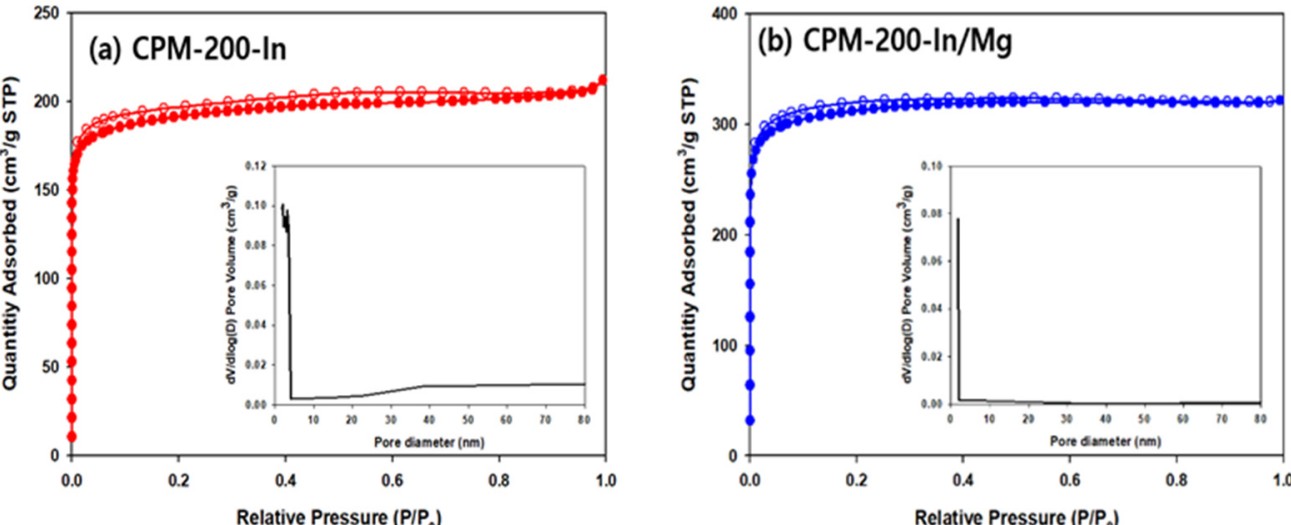

**Figure 6.** $N_2$ adsorption-desorption analysis of (**a**) CPM-200-In and (**b**) CPM-200-In/Mg.

Temperature-programmed desorption (TPD) of $NH_3$ and $CO_2$ analysis was performed to investigate acid-base nature of the catalysts. As shown in Table 2, CPM-200-In/Mg shows a higher number of basic sites, which is necessary for $CO_2$ adsorption, than CPM-200-In. $NH_3$ adsorption properties of CPM-200 catalyst was studied to explain the presence of acidic sites; these sites are necessary catalytic centers for the activation of epoxide molecules and also for the coordination with $CO_2$ molecules [8]. The number of acidic sites was almost the same in both catalysts.

**Table 2.** Number of acidic and basic sites in the catalysts.

| Catalyst | Acidic Sites $NH_3$-TPD (mmol/g) | Basic Sites $CO_2$-TPD (mmol/g) |
|---|---|---|
| CPM-200-In | 23.9 | 6.7 |
| CPM-200-In/Mg | 24.2 | 12.9 |

The $CO_2$ adsorption experiment at 298 K (Figure 7) showed that CPM-200-In/Mg reached a $CO_2$ uptake capacity of 89.5 $cm^3g^{-1}$, much higher than CPM-200-In, 49.3 $cm^3g^{-1}$ at 298 K, respectively. The increase in the basic sites in CPM-200-In/Mg is responsible to the increase of $CO_2$ uptake capacity, favoring the attraction between $CO_2$ and the catalyst.

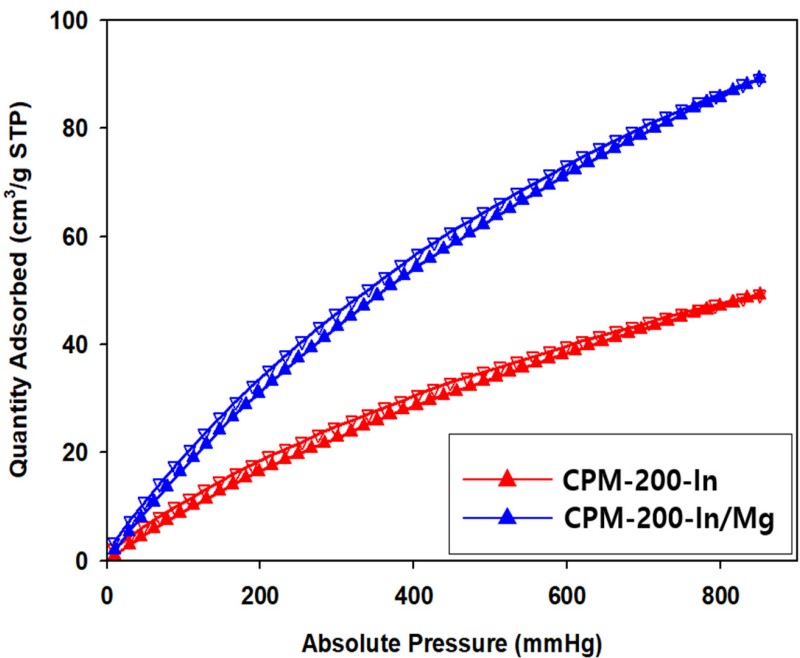

**Figure 7.** $CO_2$ adsorption-desorption analysis of CPM-200-In and CPM-200-In/Mg.

### 2.2. Cycloaddition of $CO_2$ with Epichlorohydrin

The catalytic ability of CPM-200-In/Mg was tested in the cycloaddition of $CO_2$ and epichlorohydrin (ECH). Without catalyst, the reaction did not give any ECH conversion (Table 3, entry 1). CPM-200-In/Mg and CPM-200-In alone without a co-catalyst showed very low ECH conversion at the standard condition (6 h of reaction, 38.3 mmol of ECH, 1.2 MPa of $CO_2$ pressure, 80 °C). They showed an appreciable conversion at high temperature and reaction time (110 °C and 12 h). The homogeneous catalyst precursors exhibited much lower ECH conversion (Table 3, entries 2–5). In order to promote the cycloaddition reaction, tetrabutylammonium halides (TBAX, X = Cl$^-$, Br$^-$, I$^-$) were used as co-catalysts (Table 3, entries 10, 13, 14). CPM-200-In/Mg catalyst with tetrabutylammonium bromide (TBAB) co-catalyst exhibited very high ECH conversion (90.3%) with >99% (chloromethyl)ethylene carbonate selectivity (Table 3, entry 10). Br$^-$ ions showed higher ECH conversion than iodide and chloride ions. Due to the bulky nature of iodide anion, CPM-200-In/Mg with tetrabutylammonium iodide (TBAI) exhibited lower ECH conversion (85.8%) than with TBAB because of mass transfer limitation. The catalytic performance of the CPM-200-In/Mg/TBAB system increased with the increase in temperature from 60 °C to 80 °C (Table 3, entries 10 and 12).

Various reaction parameters such as reaction time, temperature, catalyst amount, and $CO_2$ pressure are optimized for the catalytic activity of CPM-200-In/Mg/TBAB system. As illustrated in Figure 8, with increasing temperature from 40 to 80 °C, ECH conversion increased steadily; however, a small increase in ECH conversion was shown with increasing reaction time from 8 to 10 h. A maximum in ECH conversion occurred at 1.5 MPa $CO_2$ pressure, since excessive pressure decreased the concentration of ECH, disturbing the interaction between CPM-200-In/Mg/TBAB and the ECH [25,43]. The optimized reaction parameters were determined at 80 °C, $CO_2$ pressure of 1.2 MPa, after 6 h, and with 0.6 mol% of CPM-200-In/Mg with TBAB co-catalyst.

**Table 3.** Catalytic activity comparison of catalysts for the cycloaddition reaction of ECH and $CO_2$.

| Entry | Catalyst | Conversion [c] (%) | Selectivity [c] (%) |
|---|---|---|---|
| 1 | None | - | - |
| 2 | Indium salt | 3 | 96 |
| 3 | Magnesium salt | 4 | 96 |
| 4 | CPM-200-In | 3 | 98 |
| 5 | CPM-200-In/Mg | 5 | 98 |
| 6 | CPM-200-In [a] | 14 | 99 |
| 7 | CPM-200-In/Mg [a] | 17 | 99 |
| 8 | TBAB | 43.5 | >99 |
| 9 | CPM-200-In/TBAB | 81.7 | >99 |
| 10 | CPM-200-In/Mg/TBAB | 90.3 | >99 |
| 11 | CPM-200-In/TBAB [b] | 42.2 | >99 |
| 12 | CPM-200-In/Mg/TBAB [b] | 52.4 | >99 |
| 13 | CPM-200-In/Mg/TBAC | 78.6 | >99 |
| 14 | CPM-200-In/Mg/TBAI | 85.8 | >99 |

Reaction Conditions: Epichlorohydrin (ECH) = 38.3 mmol, Catalyst = 0.6 mol%, TBAB = 0.6 mol%, Pressure = 1.2 MPa $CO_2$, Temperature = 80 °C, Time = 6 h, semi-batch reaction. [a] T = 110 °C, t = 12 h. [b] T = 60 °C. [c] Determined by GC.

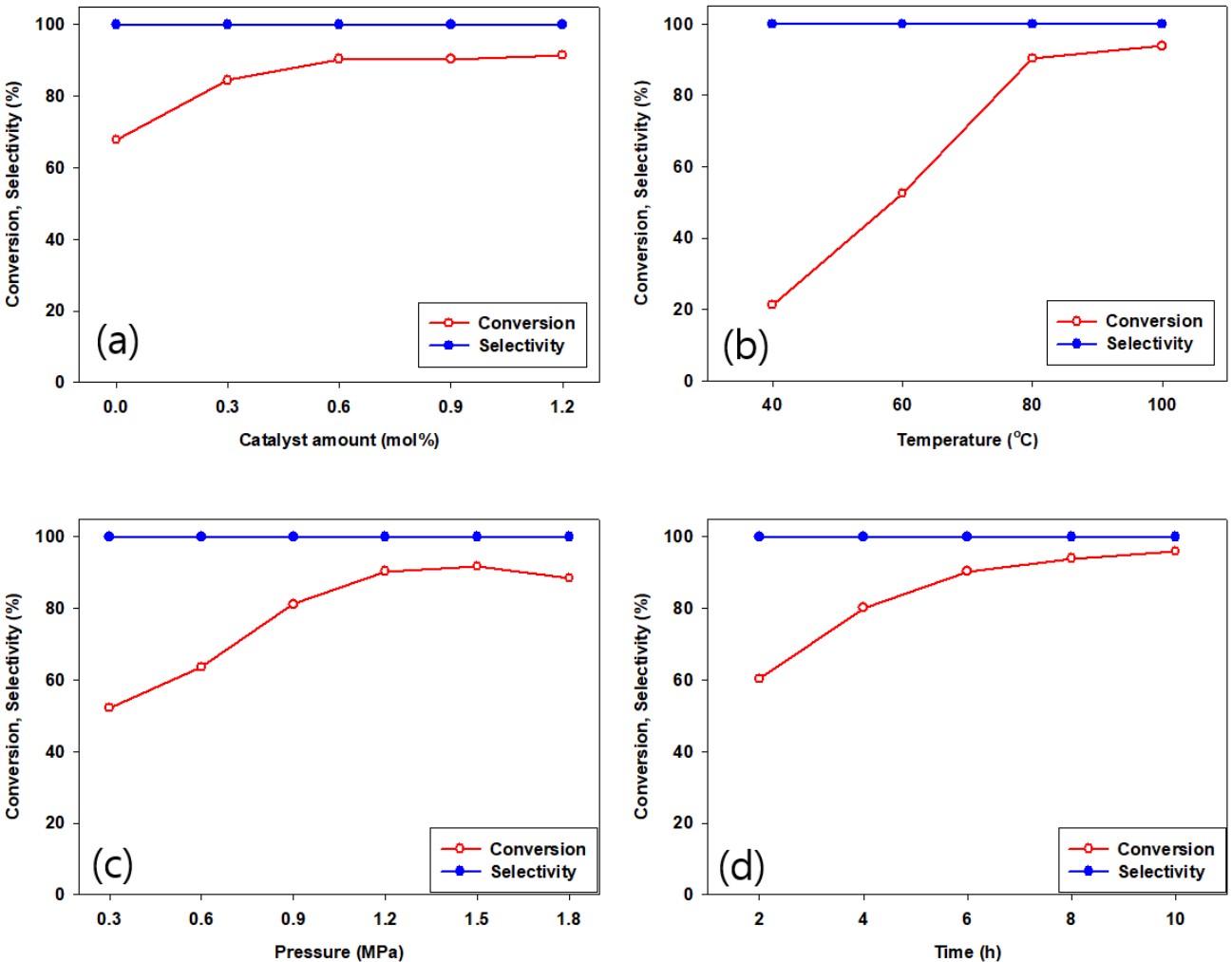

**Figure 8.** Effect of various reaction parameter on the cycloaddition of $CO_2$ and ECH with CPM-200-In/Mg catalyst (**a**) catalyst amount (80 °C, 6 h, 1.2 MPa $CO_2$), (**b**) temperature (0.6 mol%, 6 h, 1.2 MPa $CO_2$), (**c**) $CO_2$ pressure (0.6 mol%, 80 °C, 6 h), and (**d**) time (0.6 mol%, 80 °C, 1.2 MPa $CO_2$).

### 2.3. Cycloaddition of $CO_2$ with Different Epoxides

We tested various aliphatic and aromatic epoxides to investigate the versatility of the CPM-200-In/Mg catalyst. As shown in Table 4, the CPM-200-In/Mg/TBAB system was efficient with regard to various epoxides in the corresponding cyclic carbonates, except for cyclohexene oxide. High conversion of propylene oxide (87.3%), epichlorohydrin (90.3%), and allylglycidyl ether (82.1%) might be ascribed from the effective size of the epoxides, facilitating their access to the active sites. Very low conversion (14.3%) was obtained for cyclohexene oxide since the sterically hindered cyclohexene ring may prevent its approach to the Lewis acidic site [32,37,66]. In addition, it is well known that $CO_2$-based polycarbonate is easily formed from the reaction of cyclohexene oxide and $CO_2$ [67].

**Table 4.** Synthesis of cyclic carbonates from various epoxides.

| Entry | Reactant | Product | Yield [a] (%) |
|:---:|:---:|:---:|:---:|
| 1 | | | 90.3 |
| 2 | | | 87.3 |
| 3 | | | 82.1 |
| 4 | | | 71.9 |
| 5 | | | 14.3 |

Reaction Conditions: Epoxide = 38.3 mmol, Catalyst = 0.6 mol%, TBAB = 0.6 mol%, Pressure = 1.2 MPa $CO_2$, Temperature = 80 °C, Time = 6 h, semi-batch reaction. [a] Determined by GC.

### 2.4. Reusability of Catalyst

After easy separation using a centrifuge, the recyclability studies of CPM-200-In/Mg/TBAB catalyst were carried out under optimal reaction conditions (80 °C, 6 h, 0.6 mol% catalyst, and 1.2 MPa of $CO_2$ pressure). As shown in Figure 9, CPM-200-In/Mg/TBAB system can be recycled five times, without considerable loss in its initial activity. The reused CPM-200-In/Mg catalyst was analyzed by FT-IR, PXRD, and TGA to investigate its chemical and thermal stability. PXRD patterns and FT-IR spectra of the fresh and reused CPM-200-In/Mg are identical (Figures S1 and S2). The TGA of the recycled CPM-200-In/Mg catalyst showed no destruction of the original structure of the fresh one (Figure S3).

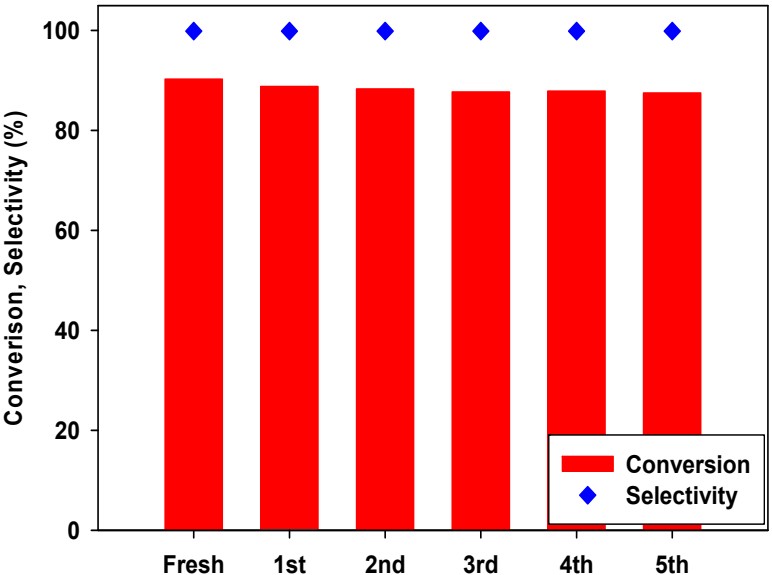

**Figure 9.** Reusability test of CPM-200-In/Mg catalyst. (epichlorohydrin (ECH) 38.3 mmol, catalyst 0.6 mol%, 80 °C, 6 h, 1.2 MPa).

### 2.5. Comparison with Other MOFs

Table 5 shows the catalytic performance of CPM-200-In/Mg together with some other reported MOF catalysts. Compared to Mg or In containing single metal MOFs (Table 5, entries 1, 8-11), CPM-200-In/Mg exhibited similar or higher cyclic carbonate TOF at lower $CO_2$ pressure and shorter reaction time. Other metal containing MOFs (Table 5, entries 2-5) in the cycloaddition of ECH and $CO_2$ are also compared with CPM-200-In/Mg (Table 5, entry 6). It also confirms superior performance of the present work.

**Table 5.** Comparison of the catalytic potential of the CPM-200-In/Mg with previously reported MOFs.

| No | MOF | Epoxide | T (°C) | $P_{CO_2}$ (MPa) | Time (h) | Catalyst Amount (mol%) | Yield (%) | TOF ($h^{-1}$) | Ref. |
|----|-----|---------|--------|------------------|----------|------------------------|-----------|----------------|------|
| 1 | Mg-MOF-74 [a] | SO | 100 | 2.0 | 4 | 3.33 | 95 | 7.1 | [55] |
| 2 | ZnMOF-1-NH$_2$ [b] | ECH | 80 | 0.8 | 8 | 1.0 | 89 | 11.1 | [68] |
| 3 | {[Zn(CHDC)(L)]·H$_2$O}n | ECH | 80 | 1.0 | 18 | 1.8 | 91 | 2.8 | [69] |
| 4 | NH$_2$-MIL-125 | ECH | 100 | 2.0 | 6 | 1.6 | 84 | 8.8 | [70] |
| 5 | *rho*-ZMOF [c] | ECH | 40 | 1.0 | 3 | 25 mg | 97 | - | [71] |
| 6 | CPM-200-In/Mg | ECH | 80 | 1.2 | 6 | 0.6 | 90 | 25.0 | This work |
| 7 | CPM-200-In/Mg | PO | 80 | 1.2 | 6 | 0.6 | 89 | 24.7 | This work |
| 8 | (Me$_2$NH$_2$)[In(SBA)$_2$] | PO | 80 | 2.0 | 24 | 0.15 | 85 | 23.6 | [72] |
| 9 | (Me$_2$NH$_2$)[In(SBA)(BDC) | PO | 80 | 2.0 | 24 | 0.15 | 89 | 24.7 | [72] |
| 10 | (Me$_2$NH$_2$) [In(SBA)(BDC NH$_2$)] | PO | 80 | 2.0 | 24 | 0.15 | 92 | 25.6 | [72] |
| 11 | (NH$_4$)$_3$[In$_3$Cl$_2$(BPDC)$_5$] | PO | 80 | 2.0 | 24 | 0.15 | 95 | 26.4 | [72] |

[a] Chlorobenzene was used as solvent. [b] ECH (20 mmol), TBAB (2.5 mol%) were used. [c] ECH (34.5 mmol), *rho*-ZMOF (25 mg), TBAB (200 mg) were used.

## 2.6. Cycloaddition Reaction Mechanism

A plausible reaction mechanism based on the experimental results and our previous DFT studies [31,33,37] is proposed, for the cycloaddition of epoxide and $CO_2$ catalyzed by CPM-200-In/Mg and co-catalyst (Scheme 2). In the first step, Lewis acidic metal centers interact with the O atom of the epoxide ring. Then, the $Br^-$ ion of the co-catalyst attacks the least hindered C atoms of epoxide, resulting in the ring opening of epoxide. Thereafter, the generated partial positive charge on the carbon atom of epoxide polarizes the $CO_2$ molecule. Finally, the cyclic carbonate is generated by the ring closure with the elimination of the bromide ion. The synergistic role of acidic sites (Lewis acidic In and Mg metal sites) and basic sites (Lewis basic motif inside MOF pores) seems essential for the improved catalytic performance of CPM-200-In/Mg.

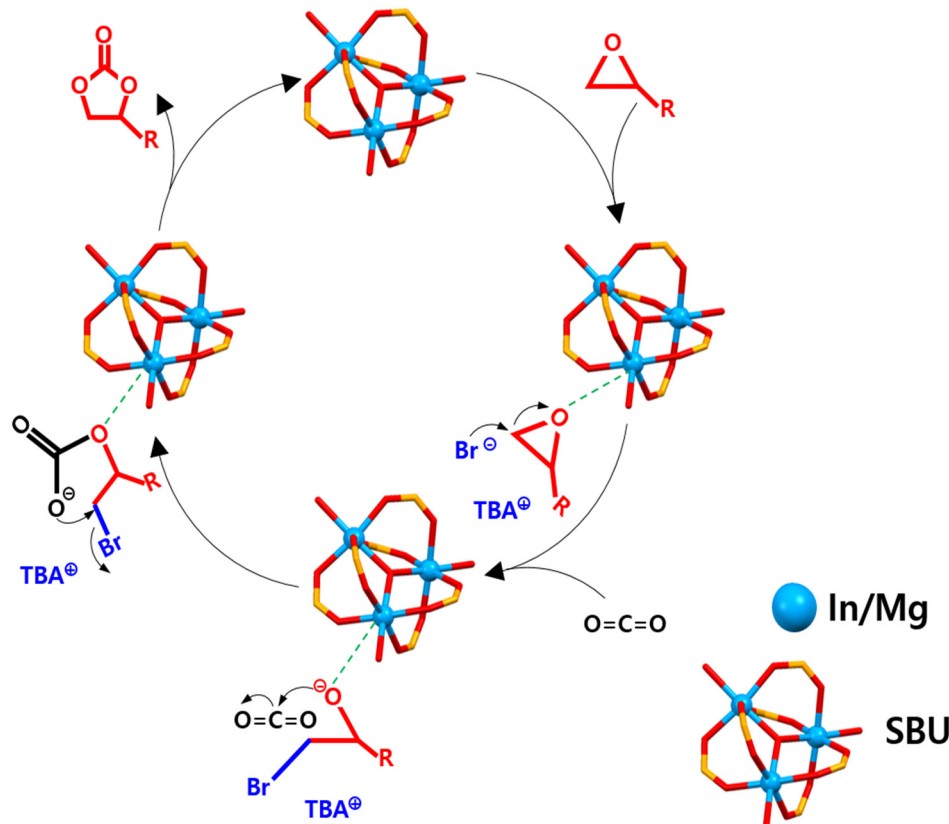

**Scheme 2.** Suggested mechanism for the cycloaddition of ECH and $CO_2$ catalyzed by CPM-200-In/Mg.

## 3. Materials and Methods

### 3.1. Chemicals

$In(NO_3)_3 \cdot xH_2O$ (99.9%), $InCl_3$ (98%), $Mg(OAc)_2 \cdot 4H_2O$ (≥99%, ReagentPlus), N,N-dimethylacetamide (DMA. 99%, ReagentPlus), acetonitrile (can, 98%, anhydrous), $HNO_3$ (65%, GR grade), HCl (37%, AR grade), ECH (≥99%, purum), and dichloromethane(≥99.9%, ACS reagent) were purchased from Sigma-Aldrich Chemical Co. (St. Louis, MO, USA) and used as received without further purification. 3,3′,5,5′-azobenzenetetracarboxylic acid ($H_4$ABTC) was prepared according to the methods previously reported [73].

### 3.2. Preparation of Catalyst

3.2.1. Preparation of CPM-200-In

CPM-200-In was prepared according to the reported method [60]. In a 20 mL glass vial, 22.0 mg of $In(NO_3)_3 \cdot xH_2O$ and 16.0 mg of $H_4$ABTC were dissolved in a mixture of 2.0 g of DMA and 1.0 g of ACN. After addition of 120.0 mg $HNO_3$, the vial was sealed and

placed in a 90 °C oven for 3 days. After cooling to room temperature, pure yellow cubic crystals were obtained.

### 3.2.2. Preparation of CPM-200-In/Mg

For the preparation of CPM-200-In/Mg in a 20 mL glass vial, 22.1 mg of $InCl_3$, 86.0 mg of $Mg(OAc)_2$, 35.6 mg of $H_4ABTC$ were dissolved in a mixture of 4.0 g of DMA and 0.8 g of $H_2O$. After addition of 120.0 mg HCl, the vial was sealed and placed in a 90 °C oven for 2 days. Pure yellow cubic crystals were obtained.

### 3.3. Cycloaddition of CO₂ and Epoxide

The cycloaddition reactions were carried out in a semi-batch reactor according to our previously reported method [43]. The detailed procedure is described in the Supplementary Materials. The products were analyzed by using a gas chromatograph (GC, Agilent technologies, HP 7890 A) with a flame ionization detector. Dichloromethane was used as an internal standard.

## 4. Conclusions

In this work, CPM-200-In and CPM-200-In/Mg MOF were successfully prepared by a solvothermal method and were characterized by using PXRD, SEM-EDS, FT-IR, TGA, XPS, BET, $CO_2$, and $NH_3$ TPD analysis. Both MOFs were used as catalysts for the $CO_2$ fixation with various epoxides, and they were shown to be highly active (TOF of 25.0/h for ECH conversion) for the cycloaddition reaction under moderate operating conditions and solvent-less conditions. Especially, CPM-200-In/Mg revealed superior catalytic performance to CPM-200-In in the cycloaddition process due to higher surface area, higher $CO_2$ adsorption capacity, and higher number of basic sites. CPM-200-In/Mg catalyst could be recycled five times, without considerable loss in its initial activity. Based on the experimental data and our previous DFT studies, a plausible reaction mechanism, including the synergistic role of the acidic sites and basic sites for the bimetallic catalyzed epoxide-$CO_2$ cycloaddition reaction, was also suggested.

**Supplementary Materials:** The following are available online at https://www.mdpi.com/article/10.3390/catal11040430/s1: experimental procedure of the cycloaddition reaction, Figure S1: PXRD patterns of reused CPM-200-In/Mg catalyst, Figure S2: FT-IR spectra of reused CPM-200-In/Mg catalyst, Figure S3: TGA curve of reused CPM-200-In/Mg catalyst.

**Author Contributions:** Conceptualization and writing original draft preparation, Y.G., writing, review, and editing, Y.C. and D.-W.P.; Supervision, D.-W.P. All authors have read and agreed to the published version of the manuscript.

**Funding:** This research was funded by National Research Foundation of Korea through Basic Research Program (2019R1I1A3A01057644).

**Data Availability Statement:** Data is contained within the article or Supplementary Materials.

**Conflicts of Interest:** The authors declare no conflict of interest.

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
