# Peer review of "Catalytic Performance of CPM-200-In/Mg in the Cycloaddition of CO2 and Epoxides"

_catalysts, doi:10.3390/catal11040430_

Round 1
Reviewer 1 Report
Below are some questions/suggestions I have for the authors that should be addressed before accepting this manuscript.
- The work lacks the novelty of chemical synthesis. Similar type of work was published recently. There is no explanation of significant novelty of the work as compared with their previous work. This should be highlighted and compared with their previous work. What are the new findings in this work and how it differs from their previous work?
- The authors simply provided so many datas; meanwhile, there is no explanation in the main text or in the supporting of the datas. Example. There are no explanations of TGA, EA, XPS, porosity study. Just a short one-sentence statement of fact that adds nothing to the discussion. The authors jut provided the spectral regions. Why the thermal stability of the materials increased in compound 4 and 5? Material CPM-200-In/Mg showed higher thermal stability than CPM-200-In, explain why this happened?. The reason should be explained.
- Chemical structures drawings are of poor quality.
- PXRD patterns of CPM-200-In/Mg after recycle is not the same as before, peaks intensities are different, this should be explained.
- The powder X-ray diffraction patterns of the prepared CPM-200 catalysts are not matched well with the simulated pattern of the CCDC, for example, there is an extra peak at 10 2Theta.
- Figure 2 FT-IR spectrum of CPM-200-In and CPM-200-In/Mg, the scale is in reverse order. The unit is “cm-1” so 500 cm-1 is greater than 4,000 cm-1. The baseline correction has gone too far and the spectrum looks like it's been clipped from top. Figure 2 should be corrected in accordance with the accepted standard.
- The catalytic results obtained by CPM-200-In/Mg/TBAB catalytic system did not fully show the high (“superior performance”) catalytic activity claimed by the authors. The catalyst is used in 0.6 mol% ratio under 1.2 MPa pressure of CO2, 80 ËšC and long reaction time = 6 h. Then, to compare with existing catalysts this should be taken into consideration and also, comparing the TOF should be much more indicative than only comparing the final conversion.
- The introduction is very concise. Not only MOF-74-Mg was used as the heterogeneous catalyst. Also there is no comparison with homogeneous systems. Some relevant references about CO2 cycloaddition reaction on MOFs should be cited, such as: Front. Energy Res. 2015, 2, No. 63, ACS Catal. 2015, 5, 6748−6752, Nat. Commun. 2015, 6, No. 5933, ACS Appl. Mater. Interfaces 2021, 13, 8344–8352.
- It is well-established that the production of cyclohexene carbonate from cyclohexene oxide and CO2 is not favored relative to aliphatic carbonates because of ring strain. Indeed, cyclohexene carbonate can be ring-opened to provide the CO2-based polycarbonate. There should be reference to these numerous studies.
- I wonder what is the purpose for authors to study NH3 adsorption property of CPM-200-In and CPM-200-In/Mg ? There is no explanation in the main text.
- In the introduction I’m missing reference and discussion of major work on cycloaddition of CO2 and epoxides, no reference to this substantial contribution to this area is given but furthermore, that research should be discussed in these context as it is highly relevant, for example: Inorg. Chem. 2018, 57, 2584–2593; Chem. – Eur. J. 2020, 26, 13686–13697; ACS Catal. 2017, 7, 3532–3539 but many others are available.
Author Response
The response is attached separatedly.

Reviewer 2 Report
Please find the attachment.

Author Response
The response is attached separatedly.

Round 2
Reviewer 1 Report
The authors improved their manuscript significantly, it is now suitable for publication and I recommend publishing this manuscript.
Author Response
Thank you for your comments.
Reviewer 2 Report
Comments have been attached.

Author Response
Thank you for your comments.
